# The Occurrence of MET Ectodomain Shedding in Oral Cancer and Its Potential Impact on the Use of Targeted Therapies

**DOI:** 10.3390/cancers14061491

**Published:** 2022-03-15

**Authors:** Maria J. De Herdt, Berdine van der Steen, Robert J. Baatenburg de Jong, Leendert H. J. Looijenga, Senada Koljenović, Jose A. Hardillo

**Affiliations:** 1Department of Otorhinolaryngology and Head and Neck Surgery, Erasmus MC, Cancer Institute, University Medical Center Rotterdam, 3015 GD Rotterdam, The Netherlands; b.vandersteen@erasmusmc.nl (B.v.d.S.); r.j.baatenburgdejong@erasmusmc.nl (R.J.B.d.J.); j.hardillo@erasmusmc.nl (J.A.H.); 2Princess Máxima Center for Pediatric Oncology, 3584 CS Utrecht, The Netherlands; L.Looijenga@prinsesmaximacentrum.nl; 3Department of Pathology, Erasmus MC, Cancer Institute, University Medical Center Rotterdam, 3015 GD Rotterdam, The Netherlands; 4Department of Pathology, Antwerp University Hospital, 2650 Edegem, Belgium; Senada.Koljenovic@uza.be

**Keywords:** oral squamous cell carcinoma, MET, receptor tyrosine kinase, ectodomain shedding, malignant potential, GPCR and EGFR signaling, biomarker, prognosis, targeted therapies (moAbs and TKIs), patient stratification

## Abstract

**Simple Summary:**

Head and neck cancer is the sixth most common cancer type worldwide, comprising tumors of the upper aero/digestive tract. Approximately 50% of these cancers originate in the oral cavity. Depending on disease stage, oral cancer patients are treated with single-modality surgery, or in combination with radiotherapy with or without chemotherapy. Despite advances in these modalities, the 5-year survival rate is merely 50%. Therefore, implementation of targeted therapies, directed against signaling molecules, has gained attention. One potential target is the MET protein, which can be present on the surface of cancer cells, orchestrating aggressive behavior. As cancer cells can shed the extracellular part of MET from their surface, it is important to identify for MET positive patients whether they possess the entire and/or only the intracellular part of the receptor to assess whether targeted therapies directed against the extracellular, intracellular, or both parts of MET need to be implemented.

**Abstract:**

The receptor tyrosine kinase MET has gained attention as a therapeutic target. Although MET immunoreactivity is associated with progressive disease, use of targeted therapies has not yet led to major survival benefits. A possible explanation is the lack of companion diagnostics (CDx) that account for proteolytic processing. During presenilin-regulated intramembrane proteolysis, MET’s ectodomain is shed into the extracellular space, which is followed by γ-secretase-mediated cleavage of the residual membranous C-terminal fragment. The resulting intracellular fragment is degraded by the proteasome, leading to downregulation of MET signaling. Conversely, a membrane-bound MET fragment lacking the ectodomain (MET-EC^-^) can confer malignant potential. Use of C- and N-terminal MET monoclonal antibodies (moAbs) has illustrated that MET-EC^-^ occurs in transmembranous C-terminal MET-positive oral squamous cell carcinoma (OSCC). Here, we propose that ectodomain shedding, resulting from G-protein-coupled receptor transactivation of epidermal growth factor receptor signaling, and/or overexpression of ADAM10/17 and/or MET, stabilizes and possibly activates MET-EC^-^ in OSCC. As MET-EC^-^ is associated with poor prognosis in OSCC, it potentially has impact on the use of targeted therapies. Therefore, MET-EC^-^ should be incorporated in the design of CDx to improve patient stratification and ultimately prolong survival. Hence, MET-EC^-^ requires further investigation seen its oncogenic and predictive properties.

## 1. Introduction: Receptor Tyrosine Kinase MET, a Suitable Target for Therapy in Oral Cancer?

Head and neck squamous cell carcinoma (HNSCC) is the sixth most common cancer worldwide and comprises epithelial cancers of the upper aero/digestive tract originating in the oral cavity, oropharynx, hypopharynx, or larynx [1]. Approximately 50% of HNSCC originates in the oral cavity [2]. Depending on disease stage, the treatment of oral squamous cell carcinoma (OSCC) patients consists of single-modality surgery, or in combination with radiotherapy with or without systemic adjuvant chemotherapy, commonly referred to as chemo–radiotherapy or chemo-radiation (CRT) [1]. Notwithstanding the advances in these modalities, the 5-year survival rate of OSCC patients is still around 50% [1,3,4]. As such, implementation of targeted therapies directed against signaling molecules known to facilitate HNSCC disease progression has gained much attention [5].

One target of interest is the predominantly epithelial receptor tyrosine kinase (RTK) MET [5,6], often upregulated in HNSCC [7]. Upon binding to its extracellular ligand, hepatocyte growth factor/scatter factor (HGF/SF) [8,9], which stimulates receptor dimerization and kinase activity [10], MET facilitates invasive growth by activating a complex network of signaling cascades, including MAPK, PI3K-Akt, STAT, and IκBα-NFκB [11,12,13]. During invasive growth, cancer cells integrate proliferation, survival, motility, and epithelial-to-mesenchymal transition [11], a program that converts epithelial cells into more mesenchymal derivatives that has emerged as a central driver of tumor malignancy [14] (Figure 1A).

In up to 25% of HNSCC, MET is aberrantly activated by mutations and amplifications [7]. Specifically, somatic mutations affecting the kinase domain are observed at a rate of around 8%, among which Y1230C, Y1235D [7]. The Y1230C and Y1235D mutations constitutively activate MET and both mutations were found in lymph node metastasis of HNSCC [17]. Interestingly, transcripts of the Y1235D mutant alleles are highly represented in regional lymph node metastasis, and barely detectable in the corresponding primary tumors, suggesting that cells carrying mutant *MET* undergo clonal expansion during HNSCC disease progression [17]. Despite a variety of scoring methods and definitions, *MET* gene amplifications have been reported to occur in 1–13% of HNSCC [7]. Although reported, the biological and clinical consequences of *MET* amplification in HNSCC need to be further investigated [7].

In the majority of cancers, including HNSCC, *MET* is transcriptionally activated by stimuli such as hypoxia, inflammatory cytokines, stromal HGF, and pro-angiogenic factors, often abundantly present in the reactive tumor-associated stroma [7,13,18,19]. In this context, MET activation occurs in already transformed cells to increase their proliferative, anti-apoptotic, and migratory potential [13].

Although MET expression is known to be associated with poor prognosis in HNSCC [20] and numerous targeted therapies are under investigation [12,21], major survival benefits have not yet been obtained [21,22]. This could be due to a lack of relevant companion diagnostics [22,23], of which development has been proven difficult for several reasons [22]. Some are of a technical nature, such as lack of reliable antibodies (Abs) and scoring systems [12,20,22], while other may be due to biology, such as proteolytic processing and more specifically ectodomain shedding [22,24].

## 2. MET’s Protein Structure and Its Degradation

The structural characterization of MET was initially resolved using COOH-terminal (C-terminal) MET antibodies [25] directed against the intracellular part of the receptor. It was discovered that the *MET* proto-oncogene [26,27,28] encodes a partially glycosylated 170 kDa single-chain intracellular precursor (p170^MET^) that undergoes terminal glycosylation and proteolytic cleavage yielding mature MET (p190^MET^) [29]. The latter is a cell-surface-associated heterodimer composed of two disulfide-linked chains. More specifically, an extracellular 50 kDa α-chain (p50^α^) and a 145 kDa transmembranous β-chain (p145^β^) possessing the intracellular tyrosine kinase domain [27,29,30,31]. 

Similar to other receptors having intrinsic kinase activity, MET is subjective to degradation, avoiding overactivity and possibly tumorigenesis [24]. Besides ligand-dependent internalization and subsequent degradation by the proteasome and lysosome [32], membranous MET is subjective to proteolytic cleavage independent of ligand stimulation [24]. In the last context, MET can be downregulated by presenilin-regulated intramembrane proteolysis (PS-RIP) [33]. During PS-RIP, the receptor is first cleaved within its juxtamembrane domain by membrane metalloproteases. This results in the shedding of a soluble MET N-terminal fragment (MET-NTF) into the extracellular space, a process referred to as ectodomain shedding. Notably, the residual 55 kDa C-terminal fragment (MET-CTF) remains anchored to the membrane [24,34,35,36]. MET-NTF shedding is driven by ADAM metalloproteases 10 and 17 (ADAM10 and ADAM17), as shown by its inhibition upon their genetic ablation [33,37,38]. Ectodomain shedding is immediately followed by a second cleavage event of the MET-CTF by the γ-secretase complex. The resulting 50 kDa intracellular domain of MET (MET-ICD) is released into the cytosol and degraded by the proteasome [33]. The MET-CTF fragments escaping γ-secretase cleavage undergo lysosomal degradation [13,16,36] (Figure 1B). 

Production of NH_2_-terminal (N-terminal) MET monoclonal antibodies (moAbs) [25,39], directed against the extracellular domain of MET, allowed further characterization of shed NTFs. Using these antibodies and breast cancer cell lines, Athauda et al. [40] specified the sizes of NTFs present in the culture medium under reducing conditions, which are 50^β^, 55^β^, 75^β^, 85^β^, and 100^β^ kDa (Table 1). Since the goal of this investigation was to evaluate whether the soluble MET ectodomain is a relevant biomarker for overall tumor burden and malignant potential (details below), no further functional analyses were performed concerning the origin of these fragments. However, 15 years earlier (1991), using stably transfected *MET* NIH-3T3 cells, Prat et al. [25] already showed that NTF p75^β^ originates from posttranslational processing of MET protein products encoded by full-length *MET* transcripts. More specifically it was shown that NTF p75^β^ is generated by proteolytic processing of membrane bound MET using wt *MET* overexpressing GTL-16 cells. Finally, treating GTL-16 cells with TPA, illustrated that generation of NTF p75^β^ from membranous p190^MET^ and p140^MET^, another C-terminal truncated transmembranous protein product of MET which is generated in the endoplasmatic reticulum [41], is mediated by protein kinase C. It was in 2010 that Schelter et al. [15] showed that DN30, one of the antibodies used by Prat et al. [25], induced ADAM10-mediated ectodomain shedding in A549 lung carcinoma cells.

## 3. MET Ectodomain Shedding in the Tumor Suppressive and Oncogenic Context

Concurrent with the view that MET proteolysis likely results from an intrinsic cellular mechanism that attenuates excessive MET signaling [13], it was shown that MET ectodomain shedding augments with increased malignant potential by taking advantage of an in vitro derived breast cancer progression model, [40]. However, Kang et al. observed that only C- and not N-terminal MET immunoreactivity was associated with poor prognosis in node-negative breast cancer. Therefore, it was suggested that the overexpression of the MET cytoplasmic tail may favor breast cancer progression upon proteolytic cleavage or by truncating mutations [44]. Accordingly, NIH-3T3 mouse embryo fibroblasts expressing a membrane-bound MET receptor fragment lacking the ectodomain (MET-EC^-^, +/−p60^β^) constitutively activated MAPK and PI3K-Akt signaling [45]. Moreover, MET-EC^-^ expression confers transforming, invasive, and tumorigenic properties to NIH-3T3 cells. Based on the previous observations, it is assumed that MET ectodomain shedding not only suppresses MET signaling by reducing the number of receptor molecules present on the membrane, but it may also give rise to membranous MET-EC^-^ that conveys invasive and aggressive properties to cancer cells, ultimately resulting in poor prognosis. 

## 4. Biochemical Evidence for MET Ectodomain Shedding in Oral Squamous Cell Carcinoma

Knowing that MET ectodomain shedding is associated with poor prognosis and possibly potentiates MET’s oncogenic potential [40,44,45], it was investigated whether this phenomenon occurs in OSCC using valid C- and N-terminal MET antibodies in parallel [42,46]. When analyzing the results of this study, it was assumed that the existence of CTFs with molecular weights reciprocal to those observed for the NTFs described by Athauda et al. [40] alongside p145^β^ under reducing conditions could be considered as evidence for the existence of MET ectodomain shedding (Table 1). First, it was established that *MET* expressing OSCC cell lines (SCC-4, SCC-25, CAL-27 and UM-SCC-14C) are subjective to ectodomain shedding by detecting CTFs (i.e., p70^β^, p90^β^, and p95^β^) under reducing conditions and detecting N-terminal MET immunoreactivity by performing an ELISA on the culture media [42]. The detection of CTFs (i.e., p95^β^, p90^β^, p70^β^, p60^β^, and p55^β^) under reducing conditions illustrated that ectodomain shedding also occurs in fresh frozen tissues of surgically removed primary OSCC [42]. Use of biological duplicates revealed inconsistencies in terms of CTF detection, suggesting that ectodomain shedding occurs heterogeneously across a tumor [42]. The existence and heterogeneity of ectodomain shedding was confirmed through comparison of C- and N-terminal membranous MET immunoreactivities on formalin-fixed paraffin-embedded whole-tissue sections, leading to the identification of regions positive for membranous C-terminal MET immunoreactivity, yet devoid of N-terminal MET immunoreactivity, which were considered to be positive for (MET-EC^-^) [42,47]. Taking everything into consideration it was concluded that MET ectodomain shedding occurs in OSCC [42,47].

To our knowledge, none of the genetic aberrations known in HNSCC lead to enhanced ectodomain shedding and stabilization of MET-EC^-^ on the membrane. Therefore, we propose a model for the molecular mechanism underlying MET ectodomain shedding below.

## 5. Crosstalk between G-Protein-Coupled and EGF Receptors Enhances MET Ectodomain Shedding

MET ectodomain shedding has been described to be regulated at the post-translational level by a crosstalk between G-protein-coupled and epidermal growth factor receptors (GPCRs and EGFRs). In 2001, using NH_2_-specific MET antibodies, Nath et al. demonstrated that lysophosphatidic acid (LPA), a serum phospholipid and GPRC ligand, and EGF, a mitogenic growth factor acting through transmembrane tyrosine kinase receptors, increase MET ectodomain shedding in A549 human lung adenocarcinoma cells. Furthermore, it was shown that both LPA- and EGF-mediated MET ectodomain shedding requires the tyrosine kinase enzymatic activity of EGFR. Additional experiments revealed that LPA mediates transactivation of EGFR through metalloprotease mediated autocrine release of HB-EGF, an EGF family growth factor, from the cell membrane and that MEK inhibitors downregulate both LPA- and EGF-induced MET ectodomain shedding, which is also facilitated by metalloproteases. These experiments suggest that in tumors earmarked by high levels of EGFR activation rates of MET ectodomain shedding increase through activation of the MAPK signaling pathway [48] (Figure 2A).

## 6. Membranous MET-EC^-^ in OSCC

Binding of gastrin-releasing peptide (GRP) to its receptor GRPR, which is a GPCR, facilitates MAPK signaling in the 1483 OSCC cell line through transactivation of EGFR. This is accomplished through metalloprotease-mediated release of TGF-α, the primary autocrine EGFR ligand in HNSCC, from the membrane. Accordingly, inhibition of metalloproteases blocks GRP-mediated EGFR tyrosine phosphorylation [50]. Although never formally proven, it is plausible that an increase in MET ectodomain shedding in OSCC results from GPCR mediated EGFR transactivation.

Increased mRNA and protein levels of the metalloproteases ADAM10 and ADAM17, responsible for MET ectodomain shedding, increase invasive behavior of OSCC and are accordingly associated with advanced tumor stages, regional lymph node metastasis, and reduced survival [51,52].

Overall, it appears that GPCR transactivation of EGFR [50] and overexpression of ADAM10/17, as observed in OSCC [51,52], result in the increased rate of MET ectodomain shedding. In combination with MET overexpression, as observed in the majority of OSCC cases [53], this results in the increased concentration of MET-EC^-^ fragments on the tumor cell membrane thus enhancing their stabilization (e.g., by dimerization or oligomerization) and constitutive activation, as also observed by Merlin et al. in NIH-3T3 cells [45] (Figure 2B). Although MET ectodomain shedding occurs in OSCC and differences between C- and N-terminal MET immunoreactivity are associated with poor survival [42,47], it has never been formally proven that a fully functional phosphorylated MET-EC^-^ exists in OSCC. Nevertheless, we hypothesize on the potential value of shedding in the stratification of patients with OSCC eligible for treatment with MET-targeted therapies.

## 7. MET-EC^-^ as Biomarker in OSCC

Accurate C- and N-terminal MET IHC analysis has allowed us to stratify resection specimens from patients diagnosed with primary and treatment-naive OSCC into three categories: MET negative (no MET immunoreactivity), decoy MET (more positivity for the N-terminal moAb), and transmembranous C-terminal MET (equal positivity for both moAbs (complete MET) or more positivity for the C-terminal moAb (complete MET and/or MET-EC^-^)) [42,46]. As OSCC patients falling in the third category developed cancers that are positive for the protein’s catalytic domain, they are likely to be eligible for MET-targeted therapies. Accordingly, an association between C-terminal MET immunoreactivity and survival was established [42]. Since a subpopulation of the C-terminal MET-positive patients show no immunoreactivity for the N-terminus, and therefore are presumed to show a suboptimal response to MET moAbs, an association between the difference between C- and N-terminal MET (MET-EC^-^) and poor survival was also established [42,47], a result that is concurrent with the tumor-promoting potential of MET-EC^-^ [45]. Hence, both C-terminal MET and MET-EC^-^ may represent useful and complementing bio-markers in the identification of OSCC patients that might benefit from MET-targeted therapies in the first line setting. To this aim, it would be recommended to use MET moAbs or tyrosine kinase inhibitors (TKIs) for cancers positive for the full-length receptor, and TKIs or a combination of both for MET-EC^-^ cancers [42,47] (Figure 3).

## 8. Discussion

Experiments using immortalized cell lines and patient-derived xenografts have illustrated that tumors harboring *MET* mutations and amplifications are responsive to MET blockade in the form of cell cycle arrest and/or apoptosis in vitro [54] and complete inhibition of tumor growth in vivo [55]. Whether there is a molecular subtype of HNSCC, and more specifically OSCC, driven by *MET* mutation and/or amplification that confers a susceptibility to targeted agents, needs to be further examined in sufficiently large and well-described patient cohorts [7]. In contrast, limited to no effects of MET-targeted therapies were observed on cell growth in wild-type *MET* tumor models [56]. Nevertheless, pharmacological inhibition of wild-type MET has negative effects on cell survival from apoptotic insults, migration, and metastasis [57]. This led to the notion that only patients whose primary tumors and/or metastases are earmarked by *MET*-specific activating (oncogenic) mutations or amplification are likely to benefit from MET-targeted therapies [11]. This also partially explains the contrasting literature reports on the success of MET-targeting agents. Yet, lack of genetic defects does not necessarily imply that MET-targeted therapies are ineffective in the wild-type setting. Several studies have shown that MET inhibitors can affect resistance to radio- and chemotherapy, and to cetuximab, a moAb directed against EGFR [53,58]. Moreover, activation of MET on dendritic cells in the TME can create a state of immunotolerance for cancer cells by downregulating the inflammatory activity of T-cells through engaging their immune checkpoints [59,60]. More precisely, similar to cancer cells, dendritic cells can suppress an immune response by expressing surface molecules, such as programmed cell death ligand 1 (PDL-1), that inhibit T-cells by binding to their membranous receptor programmed cell-death protein 1 (PD-1). As such, inhibition of MET expressed by dendritic cells might contribute to the re-initiation of the activity of T cells, enabling them to kill cancer cells. Possibly the combined use of MET-targeted therapies and immune checkpoint inhibitors, i.e., PD-1, can restore an immune competent TME resulting in the killing of cancer cells [11]. As CRT, cetuximab, and PD-1 inhibitors can be applied across different OSCC disease stages, MET remains an interesting target in the wild-type setting since its inhibition could further improve the performance of these treatment modalities [1,11].

Although MET immunoreactivity has been associated with advanced disease stage, tumor recurrence, and poor prognosis, no clear relationship has been established with response to MET-targeted therapy. Therefore, the use of MET immunoreactivity to facilitate the stratification of cancer patients into treatment categories is still a subject of debate [61]. To begin with, it has been argued that MET immunoreactivity fails as a biomarker because it does not reflect MET activity. The possibility exists that MET is highly activated in low expressing tumors, e.g., if there are tumor areas with high levels of HGF in the TME [61]. Additionally, oncogenic mutations triggering MET activation do not imply MET overexpression [62]. As MET TKIs are designed to reduce MET phosphorylation, consequently reducing its signaling activity, it is argued that levels of phosphorylated MET (phospho-MET) immunoreactivity is a far better selection criterion for the inclusion of clinical trials than MET immunoreactivity [63]. Unfortunately, use of unreliable phospho-MET antibodies led to conflicting results concerning the correlation between MET and phospho-MET immunoreactivity in non-small-cell lung cancers. Where Watermann et al. showed no correlation between MET and phospho-MET [64], Copin et al. showed phospho-MET in tumor areas that are strongly positive for MET [65]. Thus far, no MET clinical trial has used phospho-MET as a selection criterion for participation, possibly leading to the inclusion of patients that will not benefit from treatment with the investigated agent. This is not only detrimental for the patients, but may also lead to the rejection of drugs with beneficial potential [61,63]. Another limitation in the use of MET immunoreactivity to predict response to treatment, is that scoring of MET immunoreactivity is generally restricted to the membrane. Since MET remains active post-endocytosis, facilitating downstream signaling, membranous scoring leads to an underestimation of MET activity. As such, new scoring methods need to be developed that take cytoplasmic MET immunoreactivity into account to further enhance patient selection and subsequent outcomes. Finally, all of the preclinical and clinical studies that have been reported to date make use of different antibodies, scoring systems, and inclusion criteria, thus limiting the validity and reproducibility of the observed outcomes [61].

## 9. Conclusions

As MET-EC^-^ is associated with poor survival in C-terminal MET positive OSCC, it can lead to improved prediction of prognosis and a more contemplated choice on the use of type of targeted therapy (moAb and/or TKI). The fact that MET remains a target of interest in the wild-type setting [11], likely playing a role in therapy resistance, creates the need to develop guidelines concerning standardized use of MET and/or phospho-MET antibodies, scoring and evaluation in the form of companion diagnostics to improve the success of clinical trials investigating the effectiveness of MET-targeted therapies, in terms of better patient stratification and ultimately prolonged survival. It is suggested here that parallel use of C- and N-terminal MET antibodies should be incorporated in such guidelines, as it allows the identification of MET’s post-translational status, among which oncogenic MET-EC^-^, which transforming, invasive, and tumorigenic properties require further investigation in the field of OSCC.

## Figures and Tables

**Figure 1 cancers-14-01491-f001:**
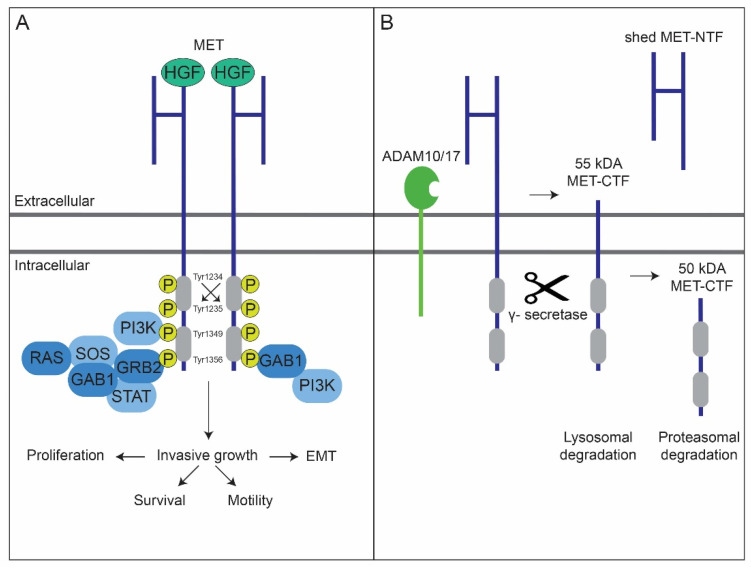
HGF/MET signaling facilitated invasive growth and presenilin-regulated intramembrane proteolysis (PS-RIP) mediated receptor downregulation. (**A**) Upon binding by its extracellular ligand, hepatocyte growth factor/scatter factor (HGF/SF), MET signaling facilitates the biological program of invasive growth by activating a complex network of signaling cascades. During invasive growth, cancer cells integrate proliferation, survival, motility, and epithelial-to-mesenchymal transition. HGF binding stimulates the kinase activity of MET through receptor dimerization and trans-phosphorylation of catalytic residues Tyr1234 and Tyr1235. Upon subsequent phosphorylation of the carboxy-terminal ‘docking’ residues Tyr1349 and Tyr1356, MET recruits a multitude of downstream signal-relay transducers. Transduction of most MET-mediated downstream signaling modules is generally regulated by the interaction with the multi-adaptor protein GAB1 (GRB2-associated-binding protein 1). Ultimately, MET and its signal transducers effectively activate downstream signaling pathways, including MAPK, PI3K-Akt, and STAT. (**B**) During presenilin-regulated intramembrane proteolysis (PS-RIP), MET is downregulated through sequential proteolytic cleavages. The receptor is first cleaved within its juxtamembrane domain by membrane metalloproteases (ADAM10/17). This results in shedding of the soluble ectodomain (MET-NTF), while a residual 55 kDa C-terminal fragment (MET-CTF) remains anchored to the membrane. The resulting 50 kDa intracellular domain of MET (MET-ICD) is released into the cytosol and degraded by the proteasome. The MET-CTF fragments escaping γ-secretase cleavage undergo lysosomal degradation. This figure and its legend, specifically 1A, was inspired by Figure 1 by Trusolino et al. [13], Figure 2 by Gherardi et al. [12], and Figure 5B by Schelter et al. [15], as well as their corresponding legends. Figure 1B and its legend, was inspired by Figure 1A and its legend by Fernandes et al. [16].

**Figure 2 cancers-14-01491-f002:**
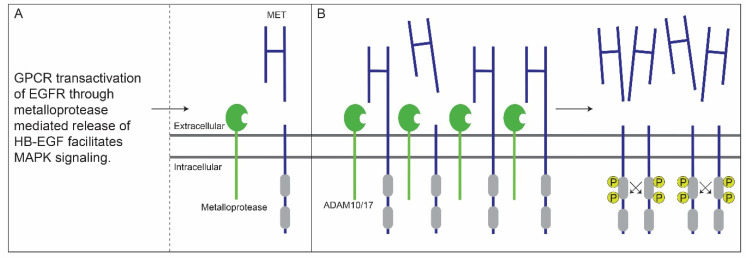
Crosstalk between G-protein-coupled and EGF receptors enhances MET ectodomain shedding. (**A**) GPCR transactivation of EGFR through metalloprotease mediated release of HB-EGF enables MAPK signaling, ultimately resulting in MET ectodomain shedding, which is also facilitated by metalloproteases. (**B**) Overexpression of ADAM10/17 and/or MET in OSCC will result in the increased concentration of MET-EC^-^ fragments on the tumor cell membrane thus facilitating stabilization (e.g., by dimerization or oligomerization) and constitutive activation of MET-EC^-^. This figure and its legend, specifically 2A, was inspired by Figure 1 and its legend by Gschwind et al. [49].

**Figure 3 cancers-14-01491-f003:**
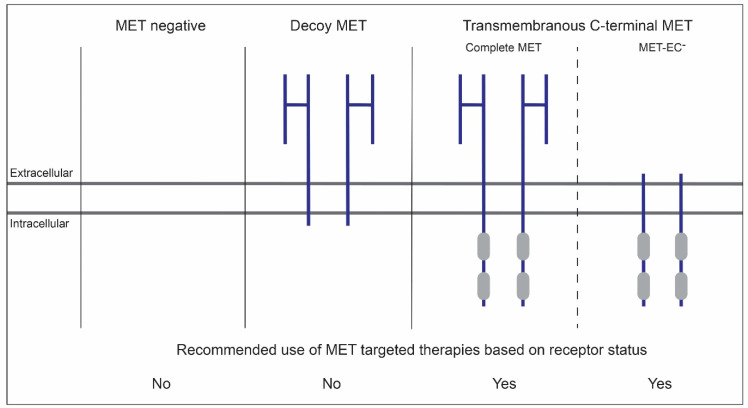
Recommended use of MET-targeted therapies, moAbs and/or TKIs, in OSCC based on the receptor’s post-translational status on the membrane. Use of MET moAbs directed against the N-term extracellular and C-terminal intracellular domain stratifies OSCCs into three categories: MET negative (no MET immunoreactivity), decoy MET (more positivity for the N-terminal moAb), and transmembranous C-terminal MET (equal positivity for both moAbs (complete MET) or more positivity for the C-terminal moAb (complete MET and/or MET-EC^-^). As patients falling in the third category develop cancers that are positive for the protein’s catalytic domain, they are presumably eligible for treatment with MET-targeted therapies. The use of MET moAbs or TKIs is recommended in case of complete MET. The use of TKIs, or a combination of TKI and moAbs, is recommended in case of complete MET and/or MET-EC^-^. This figure and its legend was inspired by Figure 10 and its legend by De Herdt et al. [42].

**Table 1 cancers-14-01491-t001:** Corresponding soluble N-terminal and presumed membranous C-terminal MET fragments resulting after ectodomain shedding named after their approximate molecular masses observed under reducing conditions [40,42].

MET N-Terminal Fragments (MET-NTFs) Observed in the Culture Medium of a Breast Cancer Progression Model [40]	MET C-Terminal Fragments (MET-CTFs) Observed in OSCC Cell Lines and Fresh Frozen Tissues [42]
p50^β^	p95^β^
p55^β^	p90^β^
p75^β^	p70^β^
p85^β^	p60^β^
* p100^β^	p55^β^

* It should be noted that this fragment should not be mistaken for the 100 kDa fragment observed by Deheuninck et al. [43], as this is a soluble extracellular fragments presumed to be the result of ectodomain shedding instead of the membrane bound extracellular domain of MET. Since the latter N-terminal fragment is able to bind HGF, however it is unable to initiate HGF signaling, due to its lacking kinase domain, it is known as the decoy receptor.

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
