# Peer review of "The Occurrence of MET Ectodomain Shedding in Oral Cancer and Its Potential Impact on the Use of Targeted Therapies"

_cancers, 2022, doi:10.3390/cancers14061491_

Round 1

Reviewer 1 Report

This a very interesting paper but the reader do not find a coherent scientific discussion of the topic announced in the title and what found in the text.

The introduction is exceedingly too long. References sometime at random or missing. Figures are not explained in the legends.

No firm evidence is provided in favor of the claim announced in the title (otherwise potentially very interesting). Technology very poor. Biochemistry data should be provided or quoted (at least western blots)

Author Response

See enclosed file.

Reviewer 2 Report

Authors clearly described tha current state of knowledge on the role of MET as a potential therapeutical target in oral cavity squamous cell carcinoma. Authors suggests that an increase in MET ectodomain shedding, stabilizes and constitutively activates membranous MET-EC- conferring transforming, invasive, and tumorigenic properties. This seems to be a promising for further clinical investigations/trials.

Author Response

Dear reviewer,

We would like to thank you for your enthusiasm and positive feedback concerning the first draft of the manuscript.

Sincerely yours,

On behalf of all co-authors,

Maria De Herdt, MSc

Reviewer 3 Report

Herdt et al. reviewed the MET protein as target therapy in head and neck cancer. They show the importance to identify MET positive patients to assess whether targeted therapies directed against the extracellular, intracellular, or both parts of MET. They propose that parallel use of C- and N-terminal MET antibodies is necessary for an accurate estimation of the presence of MET ectodomain in oral squamous cell carcinoma. Figures are nice and well explained.

Author Response

(The authors gave the same response as above.)
